# Composition of Nuts and Their Potential Health Benefits—An Overview

**DOI:** 10.3390/foods12050942

**Published:** 2023-02-23

**Authors:** Berta Gonçalves, Teresa Pinto, Alfredo Aires, Maria Cristina Morais, Eunice Bacelar, Rosário Anjos, Jorge Ferreira-Cardoso, Ivo Oliveira, Alice Vilela, Fernanda Cosme

**Affiliations:** 1CITAB, Centre for the Research and Technology of Agro-Environmental and Biological Sciences, Inov4Agro, Institute for Innovation, Capacity Building and Sustainability of Agri-Food Production, University of Trás-of-Montes and Alto Douro, Quinta de Prados, 5000-801 Vila Real, Portugal; 2CQ-VR, Chemistry Research Centre—Vila Real, University of Trás-os-Montes and Alto Douro, Quinta de Prados, 5000-801 Vila Real, Portugal

**Keywords:** antioxidant activity, consumer perception, fatty acids, fiber, health benefits, minerals, phenolic compounds, vitamins, volatile compounds

## Abstract

The possibility that nut intake may defend human health is an interesting point of view and has been investigated worldwide. Consequently, nuts are commonly promoted as healthy. In recent decades, the number of investigations proposing a correlation between nut consumption and a decrease in the risk of key chronic diseases has continued to increase. Nuts are a source of intake of fiber, and dietary fiber is associated with a reduced occurrence of obesity and cardiovascular diseases. Nuts likewise provide minerals and vitamins to the diet and supply phytochemicals that function as antioxidant, anti-inflammatory, and phytoestrogens agents and other protective mechanisms. Therefore, the main goal of this overview is to summarize current information and to describe the utmost new investigation concerning the health benefits of certain nuts.

## 1. Introduction

Currently, consumers are concerned about making a diversified and well-balanced diet. Therefore, the inclusion of nuts in the diet has undergone significant increases due to a growing recognition of their unique nutritional value, distinctive taste, flavor, nutraceutical properties, and healthy bioactive compounds, including high-quality proteins, fibers, minerals, tocopherols, phytosterols, and phenolic compounds [1]. Nuts are usually described as dry fruits with an edible seed and a hard shell, with cashews (*Anacardium occidentale*), walnuts (*Juglans regia*), almonds (*Prunus dulcis*), chestnuts (*Castanea sativa*), pistachios (*Pistacia vera*), and hazelnuts (*Corylus avellana*) as the ones with higher production worldwide [2]. There is the recognition that nuts are a good source of many nutrients, including monounsaturated and polyunsaturated fatty acid profile, vitamins E and K, selected minerals such as magnesium, copper, potassium, and selenium, dietary fibers, carotenoids, and phytosterols with potential antioxidant action [3]. In addition, the ease of transport due to their size makes them even more recommended to be consumed in all situations. In addition, the ingesting of nuts is often related to reducing risk factors for chronic diseases, due to the fatty acid profiles, squalene, fibers, vegetable proteins, minerals, vitamins, carotenoids, and phytosterols with potential antioxidant action [4]. Curiously, in all nuts, most of the antioxidants are located in the pellicle, as shown for almonds [5,6] and peanuts [7], and they are lost when the skin is removed [8]. In addition, in pistachios, most of the antioxidants are destroyed when the hard shells are cracked [9]. This review article aims to synthesize the current state of knowledge on the nutritional composition and health outcomes of some selected nuts.

## 2. Nuts

### 2.1. Proteins

Nuts are a rich source of proteins and essential amino acids as indicated by the USDA National Nutrient Database for Standard Reference [10] and as presented in Table 1. The major sources of proteins are peanuts, almonds, and pistachios, while chestnuts are the poorest in proteins. Chung et al. [11] reported higher protein content for some of these nuts, which can be ascribed to different geographic regions. The protein content also varied within the same nut species, denoting a significant effect of cultivar [12,13,14,15]. Other factors, such as the harvest year, post-harvest storage, and even the processing method [16] can affect the content of proteins in nuts. For example, Dodevbka et al. [17] reported differences between raw, boiled, and roasted nut samples from Serbia. The seed storage proteins are the main type of proteins present in nuts and are responsible for nut allergies [18]. Except for chestnuts, the other nine nuts referred to in Table 1 are the most common nuts capable of triggering adverse allergic reactions in some people. The proteins involved in nut allergy belong to different families, especially 2S albumins, globulins (legumins and vicilins), non-specific lipid transfer proteins (nsLTP), plant pathogenesis-related proteins (PR-10), profilins and oleosins [18]. Amandin, a legumin-type protein, is the most abundant protein in almonds, while the PR-10 Cor a 1 is the main allergen in hazelnuts, and the legumin Ju r 4 is the prevalent allergen in walnuts. The 2S albumins Car I 1 and Pis v 1 are dominant in pecan nuts and pistachios, respectively. The vicilins Ana o 1, Ara h 1, and Mac Ii1 are the most common allergens in cashew nuts, peanuts, and macadamia nuts, respectively [18].

Regarding the amino acid profile of each nut (Table 1), there is a considerable variation in the content of essential and non-essential amino acids. The nut protein composition is dominated by hydrophobic amino acids, followed by acidic, basic, and hydrophilic amino acids [19]. Among the non-essential amino acids, glutamic acid is the most important, ranging from 0.02 g/100 g in chestnuts to 6.21 g/100 g in almonds. The second major non-essential amino acid is arginine ranging from 0.12 g/100 g in chestnuts to 3.08 g/100 g in peanuts, followed by aspartic acid that ranges between 0.03 g/100 g in chestnuts and 3.15 g/100 g in peanuts. Leucine is the most essential amino acid, followed by phenylalanine and valine. Chestnuts present the lowest values of these essential amino acids (0.10, 0.07, and 0.09 g/100 g for leucine, phenylalanine and valine, respectively), while peanuts are the richest source of leucine and phenylalanine, and pistachios are the richest source of valine. Although the amino acid profile can differ significantly with variety and location, studies with 23 hazelnuts in northeast China [20] revealed the dominance of the same non-essential and essential amino acids (glutamic acid, arginine, aspartic acid, and leucine) described in Table 1 for hazelnut. The composition and dominance of essential and non-essential amino acids can influence several attributes of nuts, including the taste, aroma, or color being used, for example, for the characterization of almond cultivars [21].

The essential amino acid contents and their digestibility determine the nutritional value of a food protein. Although nut proteins are often recognized as incomplete proteins (i.e., do not contain all essential amino acids) when compared to animal proteins, their consumption is strongly associated with cardiovascular health [22]. Moreover, the presence of large quantities of arginine in all tree nuts has positive effects on immune response and inflammation, and cardiovascular function, including its key role in reducing the risk of cardiovascular disease and reproductive performance [23]. The health benefits of nut consumption can be enhanced by combining different protein sources to provide adequate levels of all essential amino acids.

### 2.2. Vitamins

Vitamins are essential for a balanced and healthy diet. Nuts contain fat-soluble vitamins (ascorbic acid, B1, B2, B3, B6) and antioxidants such as α-tocopherol (vitamin E), promoting better health, playing an important role against the aging process, improving brain function, and helping consumers to have healthy skin [24,25]. According to studies carried out by several researchers, the existence of vitamin C (ascorbic acid) is an important antioxidant for human colon cells [26,27]. The nut’s nutritional value depends on its chemical composition, and this is the result of the interaction of the cultivar (genotype), meteorological factors such as temperature and radiation, and production practices [28,29,30]. As Table 2 shows, walnuts, almonds, pine nuts, and hazelnuts are especially rich in vitamin E. Almonds, cashews, pistachios, walnuts, and peanuts are abundant sources of B vitamins. The concentration of folic acid was higher in pistachios and chestnuts. It is also the chestnuts that reveal the highest amount of vitamin C.

### 2.3. Minerals

Nuts are also rich sources of minerals such as magnesium and potassium (Table 3). In recent years, increased consumption of nuts has been considered good for human health to increase the intake of certain minerals, and they are considered a heart-healthy snacks when eaten in moderation [34]. Nuts are an important food source of minerals such as copper and magnesium. These two minerals may be protective against coronary heart disease. Nuts are also fairly high in potassium, particularly pistachio and cashew nuts (Table 1). Most nuts have a decent amount of zinc and iron, but pine nuts, cashews, and almonds stand above the rest. In contrast, nuts do not have a high content of calcium, still, some nuts such as almonds are better in terms of calcium content.

### 2.4. Fiber

Fiber is a health-promoting nut ingredient. The intake of dietary fiber is inversely related to obesity, type two diabetes, cancer, and cardiovascular disease according to epidemiological and clinical studies [38]. Among nuts, almonds present the highest content of fiber (Table 4), with a clear effect of genotype influencing its amount recorded [39]. Some works have highlighted the influence of genotype on the fiber content found in almonds [40], ranging from 6.88% to 9.74% in blanched almonds [41], showing that almond skin is also responsible for the fiber content of this nut, as it is composed of around 60% of fiber [42]. However, not all available data follow the same trend, with similar values of fiber recorded for different cultivars [43,44,45]. Cashews have the lowest fiber content among the referred nuts, with recent works pointing at values always around 3% to 4% [46], with no apparent significant effect of the cultivar on its content, although comprehensive studies are lacking for this specific nut. Chestnuts are considered to be a good source of dietary fiber [47], with similar values to those of cashews. The content of fiber in chestnuts has been the subject of studies that cannot find a trend on the factors behind their variation. Some authors point out the clear effect of cultivar on fiber content [43,44,48,49] or area of production [50] or year [43,44]. However, other works clearly state the similar content of fiber, regardless of cultivar [51,52,53]. The hazelnut fiber content is usually referred to as ranging from 6.5 g/100 g to 9.7 g/100 g [54]. Researchers have found higher amounts of fiber in some cultivars, such as the Turkish tombul hazelnut (12.9 g/100 g) [54], or other cultivars, with fiber values ranging from 9.8 g/100 g to 13.2 g/100 g (dry weight basis), being lowest in Tonda di Giffoni and highest in Campanica [55]. This also shows the variation of fiber content among cultivars, also recorded in the comparison of sixteen hazelnut cultivars [56]. For pistachio, the available works dealing with fiber content are few. However, early data indicate a content of 1.1–2.0% [57,58] although more recent works show considerably higher values. Dreher [59], Bulló et al. [60], and Terzo et al. [61] refer to values of fiber as around 10%, with Rabadán et al. [39] suggesting that the major factor between variations is the crop year and related weather conditions. Finally, walnut presents an intermediate amount of fiber when compared to other nuts. Although the majority of available works indicate values ranging from 4% to 6% [59,62,63,64,65], some authors have found considerably different amounts of fiber, namely Özcan [66], which indicates 1.8%, and Özcan et al. [67] that reports values between 3.8% and 3.9%. Again, the major factor affecting the fiber content of walnut is the genotype, with a slight effect also found to be caused by the crop year and related weather conditions [39].

### 2.5. Lipids and Fatty Acids

Nuts are rich in several nutrients, although with great differences between them and minor but sometimes still significant variations within cultivars. Lipid content and fatty acid profile are two of the parameters that can change considerably when discussing nut composition (Table 4 and Table 5). Besides these great variations between nut species, changes in lipid content and profile can also occur due to several other factors, with genotype as one of the most important that influences nut composition. Recent works show that genotype and the environment are key factors behind changes in several compositional parameters of some nuts, namely fat content [39]. There are some very good examples in the available literature, and to illustrate this fact, we will refer only to some for each nut.

For almonds, Summo et al. [69], working with samples from a germplasm collection under the same growing condition, recorded variations of lipid content, depending on the cultivar, from 42.4% to 56.2% (fresh weight). Barreca et al. [70] also reported a significant cultivar effect on the content of lipids in almonds. Almonds are also known for their interesting fatty acid profile, which is mainly composed of monounsaturated (MUFA) (60%) and polyunsaturated (PUFA) (30%) fatty acids, with a predominance of oleic, linoleic, palmitic, or stearic acids [43,44,71,72]. The work of Summo et al. [69] also shows the effect of the genotype on the fatty acid profile. Although major fatty acids are the same across the studied cultivars, changes can be observed in the individual amount of each fatty acid, as well as for the sum of unsaturated (mono- or polyunsaturated) and saturated (SFA) fractions. For cashew nuts, recent studies show great variability in fat content and associated fatty acid profiles when comparing different production regions. The work of Rico et al. [46], analyzing 11 cashew origins, shows that fat content can vary from 45.05 g/100 g in Vietnamese samples to 50.40 g/100 g in samples from Kenya. In the fatty acid profiles, oleic, linoleic, and palmitic acids are the three major ones. Although monounsaturated fatty acids represent the major fraction in all samples, followed by saturated fatty acids, at least in one sample, the second most important fraction is polyunsaturated fatty acids.

Chestnuts are featured with low-fat content and compared to other nuts, such as hazelnut, macadamia, pecan, or almond, chestnuts, exhibit the lowest fat content [73]. However, in this minor chestnut fraction, fat-soluble bioactive compounds, such as tocols and phytosterols, are present in higher quantities when compared to fat-rich nuts. They contain a high quantity of essential fatty acids (those that must be provided by food intake, as they are not synthesized in the body but are necessary for health) [74,75,76], either saturated or unsaturated, linked to several processes involved in health and chronic diseases [77]. Among them, the most important unsaturated fatty acids are linoleic and linolenic acids [75,78]. Fat content and fatty acid profiles can, as for other nuts, change significantly among cultivars. A thorough study of 17 chestnut cultivars produced in Portugal shows significant variations ranging in fat content from 1.67% to 3.50% [76]. Chestnut fat is primarily composed of three fatty acids, namely linoleic, oleic, and palmitic acids, with a predominance of polyunsaturated fatty acids. However, when comparing samples, significant variations of these fractions can be seen, with some presenting almost the same amount of mono- and polyunsaturated fatty acids. Similarly, the amount of saturated fatty acid also recorded significant variations across cultivars.

Among nuts, hazelnut presents one of the highest contents of fat, above 60% (Table 5), with some authors indicating the amount of fat above 70%, depending on the cultivar [79] or even on the canopy position of the fruits [80]. The fat present in hazelnuts is mainly composed of MUFA, representing around 80% of the total fatty acid content, and oleic acid is the major individual monounsaturated fatty acid [81,82,83]. Polyunsaturated fatty acids represent the second major fraction in hazelnut fat, almost exclusively due to the content of linoleic acid [84,85]. However, some works have found that SFA can represent the second major group of fatty acids [66,69], influenced by the higher content of palmitic acid.

Like most other nuts, pistachio is rich in fat, the available works indicating values around 50% [86,87,88,89], although some cultivars can have increased fat content, reaching values as high as 74.15% [90]. Following the trend of other nuts, pistachio fat is rich in unsaturated fatty acids, namely MUFA. This fraction is mainly composed of oleic acid, with a contribution from palmitoleic acid, while the second most important fraction, PUFA, is mainly composed of linoleic acid [83,89,91]. Regarding SFA, the minor fatty acid fraction is made almost entirely of palmitic acid [82,86].

The fat content of walnut is very high, with average values that can be surpassed only by hazelnuts [82]. Although the fat content is in the 60% range, considerable variations have been observed when comparing cultivars. Values varied between 49% [92] and 82% [93]. However, as referred before, most of the works show values of fat around 60%, with some variations associated with the studied cultivar [94,95]. Walnut fat is mostly composed of unsaturated fatty acid, namely PUFA, while MUFA is the second most important type of fatty acid [96,97,98]. Linoleic and linolenic acids are the ones responsible for the high amount of PUFA, with oleic as the major MUFA. Regarding SFA content, palmitic and stearic are the ones present in higher amounts [92,94,95] (Table 5).

### 2.6. Phenolic Compounds

Like in numerous other crops, phenolics are present in nuts. Many studies are reporting the beneficial effects of nut consumption on human health, including cardioprotective, neuroprotective, antidiabetic, anti-inflammatory, and antioxidant properties [99,100,101,102,103]. Studies have shown that the consumption of nuts improves flood lipoprotein profile [104,105] and gut microbiota [106]. These health effects are mainly due to the presence of several type of compounds, including phenolics, as reported by Lamuel-Raventos and Onge [102]. Each nut species presents its typical phenolic profile and content. For example, Liu et al. [107] found a high content of phenolics, such as vanillic acid, catechin, naringin, quercetin, and ellagic acid, in chestnuts, while Smeriglio et al. [108], in almonds, found a high content of phenolics, such as quercetin, kaempferol, and isorhamnetin. Instead, Taş and Gökmen [109] reported high levels of procyanidins A and B, trimers and tetramers, and prodelphinidin in peanuts. Table 6 shows several examples of phenolics found in the most common species of edible nuts. Similar to other crops, the variation in both profile and content of phenolic of nuts is highly related to genotype, cultural practices, climate conditions, fruit ripeness stage, storage, and post-harvest settings [110,111,112,113,114,115]. In addition, differences in the methods used to extract and quantify phenolic compounds (e.g., microwave-assisted extraction—MAE; supercritical CO_2_ extraction—SC-CO_2_; enzyme-assisted extraction—EAE; pressurized liquid extraction—PLE) by researchers may interfere with the number of phenolic compounds identified. However, based on the literature, it is possible to find a more or less common pattern.

The most abundant phenolics in almonds are catechin, epicatechin, protocatechuic acid, ferulic acid, kaempferol, and isorhamnetin [108,117]; in chestnuts are gallic acid, vanillic acid, syringic acid, catechin, and ellagic acid [119]; while in hazelnuts, the preponderance is for the catechin, epicatechin gallate, and gallic acid [121,122]; in peanuts, *p*-hydroxybenzoic acid, *p*-coumaric acid, ferulic acid, and epicatechin dominate [124]; in pistachios, gallic acid, syringic acid, catechin, and epicatechin [126]; while pecans and walnuts have in common high contents of chlorogenic, caffeic, *p*-coumaric, ferulic, ellagic and syringic acids [118]. In general, all nuts have in common the presence of high amounts of phenolic acids and flavonoids. The anthocyanins are present in vestigial amounts and are therefore not considered.

All these compounds are highly important because they have been associated with important beneficial effects on human health, as reported in the review of Lamuel-Raventos and Onge [102] and De Souza et al. [103]. Consumer perception of their beneficial effects has increased the intake of nuts. Different important findings from researchers have also contributed to the increment of such products in the human diet. For example, Brown et al. [130] found that higher nut consumption was associated with a reduced prevalence of high cholesterol and blood pressure, diabetes, and gallstones, due to the richness of phenolic compounds. In addition, Musarra-Pizzo et al. [131] tested a mix of phenolics present in natural almond skin and found that epicatechin and catechin were able to stop the growth of *Staphylococcus aureus*, suggesting that extracts from almond skins can be used to develop novel products for topical use. Neuroprotective effects against Alzheimer’s disease were found in almonds, hazelnuts, and walnuts due to their richness in tocopherols and phenolics [132].

### 2.7. Aroma and Flavor Compounds

The aroma compound profile of nuts is dependent on geographical origin and thermal processing and the presence of microorganisms. In almonds, several studies indicate aldehydes as the major volatiles, namely benzaldehyde [133,134,135] with a characteristic bitter-almond flavor, although this compound might not be found in several cultivars [136,137]. Besides terpenoids and substances derived from amino acids, volatiles are usually present as a result of the oxidation of fatty acids [138].

Processing causes several modifications, either in the number of compounds, but also in the chemical classes present [136,137]. In the work of Elmore et al. [139], they verified that walnuts from China and Ukraine contained high levels of lipid-derived volatiles from the linoleic acid breakdown (hexanal, pentanal, 1-hexanol, and 1-pentanol) and α-linolenic acid breakdown (1-penten-3-ol), whereas Chilean walnuts contained high levels of alkylbenzenes. Pyrazines are the major group of aromatic compounds in peanuts. They are formed by the thermally induced the Maillard reaction. The same applies to other nuts, such as pistachio and hazelnut. It is the roasting process that makes the fruit commercially viable and valuable, improving the nut’s sales and sensory characteristics [86]. Two pyrazines represent peanut flavor: 2,5-dimethyl pyrazine (with a characteristic nutty aroma) and 2-methoxy-5-methyl pyrazine (roasted nutty aroma) (Figure 1).

In hazelnuts, the results from Kiefl and Schieberle [140] showed that the aroma-active compounds 2-acetyl-1-pyrroline, 2-propionyl-1-pyrroline, 5-methyl-(*E*)-2-hepten-4-one (fibertone), 2,3-diethyl-5-methyl pyrazine, 3,5-dimethyl-2-ethyl pyrazine, and 2-furfurylthiol are appropriate odorant indicators to distinguish the several nut aromas. Specifically, the roasted or nutty aroma of roasted hazelnuts was developed if both 5-methyl-(*E*)-2-hepten-4-one and 3-methyl-4-heptanone were higher than 450 μg/kg, whereas the sum of the two 2-acyl-1-pyrrolines and two pyrazines should not exceed 400 μg/kg to avoid an over-roasted odor. A favored aroma can be obtained for each cultivar if specific temperatures, roasting techniques, and roasting times can be applied.

One major quality concern related to nuts is the development of off-flavors due to the formation of oxidative degradation products [141,142]. Various volatiles are involved in off-flavor; 1-Pentanol, 1-hexanol, and hexanal are the most important volatiles involved in off-flavor, and their presence at the highest levels is a synonym of nut degradation.

## 3. Impact of Nuts Processing on Nutrients and Phytochemicals

The phytochemicals in tree nuts have been linked to various health benefits, but processing steps can affect their bioavailability. Nuts can be processed in various ways to create different final products. For example, nuts that are consumed are often dehulled, peeled, blanched, and roasted [143,144,145]. Roasting is a common processing method used to preserve the quality and storability of nuts. It improves the flavor, aroma, color, texture, and appearance of the nuts through non-enzymatic reactions, such as Maillard browning. Roasting also inactivates enzymes that accelerate nutrient deterioration, remove microorganisms and food contaminants, and reduce degradative reactions such as lipid oxidation and rancidity, which are major factors that limit the shelf life of nuts. Additionally, the roasting process alters the microstructure and chemical composition of nuts, resulting in changes such as moisture reduction, modifications to lipids, changes in color, and the development of unique roasted flavors through the Maillard reaction [146,147]. Thus, the roasting process improves the nuts’ sensory characteristics such as flavor, color, taste, texture, appearance, and crispiness [145]. This improves the overall sensory characteristics of the nuts, making them more appealing to consume. The antioxidant activity, nutritional content, and total phenolic compounds in nuts may decrease after blanching and peeling, but roasting can improve these factors by releasing bound phenolic compounds and forming Maillard reaction products such as melanins [109,147]. However, the research on the effect of roasting on the phenolic compounds in nuts is limited. Based on available studies, the impact of roasting on the phenolic compounds in nuts can vary depending on the roasting temperature and duration. Some studies indicate that lower temperatures or shorter heating times may increase phenolic compounds, but higher temperatures or longer heating periods may decrease phenolic compounds [148]. For example, in hazelnuts, the content of flavan-3-ols (catechin and epicatechin) decreases significantly when roasted, with significant differences observed between raw nuts with skin and roasted nuts without skin. Thermal treatment also negatively impacts the content of procyanidin dimers and trimers in hazelnuts. Studies have shown that polyphenols in hazelnuts are mostly present in the skin and that roasting reduces the levels of phenolic compounds in most nuts, not only because of the removal of the skin but also due to the chemical degradation of many phenolic compounds [149]. These compounds are highly unstable and may be lost during processing, particularly when heat treatment is involved. Roasting can also alter the levels of antioxidants in the nuts, as the level of individual phenolics is higher in whole unroasted nuts [150] and alters the protein profile and allergenic properties [151]. Previous studies suggest that roasting enhances the allergenicity of roasted peanuts compared to raw peanuts [152], but the same was not observed in almonds [153]. The antioxidant activity of raw and roasted nuts depends on the type of nut and the roasting conditions. According to Schlörmann et al. [154], roasting can lead to a decrease in antioxidant activity in some nuts (hazelnut and walnut), but in others (almond and pistachio), the activity remains stable or is slightly enhanced. This decrease in activity is due to the loss of polyphenols due to thermal treatment, but the formation of antioxidant-active compounds due to Maillard reactions can counter this effect. The impact of roasting on bioavailability is still uncertain and requires further research. It is also important to evaluate the necessity of thermal processing by proving that nutritional and other properties are of great value, with antinutrients considerably decreased [155]. Thermal processing significantly reduces the protein, ash, and fiber content. The decrease in protein content may be caused by high-temperature denaturation and/or solubilization [156]. Additionally, the precipitation of mineral components leads to a decrease in ash content. In terms of carbohydrate content, roasting can increase it from 4.17% to 5.5%. This may be due to the hydrolysis of carbohydrates and to the reduction of other compounds in nuts due to thermal processing, making them easier to capture [109].

Concerning the effect of hot water blanching on protein composition, the results also depend on species and conditions of thermal processing [18]. Tian et al. [157] demonstrated that subjecting peanuts to 100 °C for 20 min reduced their allergenicity, due to the denaturation of allergenic proteins and to the transition of low molecular weight to the boiling water [158]. In turn, boiling almonds for 10 min [153] or cashews and pistachios for 60 min did not affect their properties [159].

## 4. Nut Consumer Perceptions of Health Benefits

Plant science research has been primarily focused on increasing production, with health benefits as a minor concern. The food industry is currently adapting its market trends to accommodate sustainability values, especially those related to health benefits, as they are increasingly researched by consumers [160], based, on nuts, on the phytonutrients present in these foods [161]. The current use of phytonutrients by food producers and the knowledge of their effect on the prevention of chronic disease points out the need for a careful look at crop production strategies (fertilization, season, soil fertility, and irrigation) affecting the quantitative and qualitative profiles of these compounds, but also to post-harvest techniques (processing or packaging) that can modify phytonutrients [161]. There is mounting evidence of the potential health benefits of a nut-rich diet. The ingestion of phytochemicals from nuts and their positive influence on several diseases (cancer, heart disease, stroke, hypertension, birth defects, cataracts, diabetes, diverticulosis, and obesity) are established [68,162,163,164]. There are many phytochemicals present in nuts that can be responsible for their health-promoting activities. Of those, one must refer to the vitamins, carotenoids, phenolic acids, or flavonoids, and their role in the prevention of certain cancers and cardiovascular diseases, but also to phytoestrogens, organosulfur compounds, fiber, or isothiocyanates (reviewed by several authors [68,165,166]). Nuts have been traditionally looked at as a high-fat and high-calorie food that should be consumed in moderation, which may be part of the reason why their intake is still below the recommended amount [167,168,169]. Although the link between weight gain and nut intake has been disproven [170], the usual high cost of nuts is another barrier to the increase in daily intake by consumers. The intake of nuts has been linked to several benefits to health, including favorable plasma lipid profiles, reduced risk of coronary heart disease, certain types of cancer, stroke, atherosclerosis, type-2 diabetes, inflammation, and several other chronic diseases [68,171,172]. However, it appears that consumers are not fully aware of the potential benefits of the intake of nuts. Recent works have shown that consumers link nuts to the high content of fat and proteins and that they are healthy. Nevertheless, a large percentage of consumers are still not aware of the link between nuts and the effects on blood cholesterol, cardiovascular disease risk, obesity, cancer, or diabetes [173,174,175,176,177,178].

Recently, there has been a huge effort to emphasize the beneficial action for the health by changing consumers’ eating habits, leading them to increase the consumption of certain foods such as nuts. There is no doubt that an informed consumer makes better decisions when choosing certain foods. In the long term, a higher intake of nuts will lead to clear benefits in the health sector, but other sectors will also benefit, such as producers and sellers.

Major concerns of the food industry related to the production and commercialization of nuts are the effects of processing and storage on the quality of nuts. Both temperature and humidity after harvesting can influence the appearance, moisture content, texture, and sensory characteristics of nuts [179]. Specifically, higher post-harvest temperature conditions can reduce crispness, increase moisture content and change oiliness and sweetness, resulting in the development of rancidity [180]. According to Mexis et al. [181], the alteration of sensorial characteristics leads to the formation of unpleasant flavors in pistachios, almonds, peanuts, and walnuts, as a result of alterations in the oxidation rate caused by high storage temperatures. It is mentioned in this study that storage temperatures of 30, 36, and 40 °C showed that nuts are more rancid compared to those stored at 8, 10, 20, or 25 °C.

Another possible alternative to increase the shelf life of nuts is the use of suitable packaging to reduce the problems mentioned above. Food can be packaged properly using modified atmosphere packaging or vacuum packaging to control the oxidation reaction [182]. The packaging material will be an important aspect to take into account as it will influence the shelf life of the nuts, will affect the respiration and transpiration rates of the fruit, as well as, the development of microorganisms. Fernandes et al. [182], in their comparative studies of chestnut conservation packages, concluded that chestnut conservation through the use of a specific packaging can have a substantial impact on preserving the color and texture of the fruit, preventing loss of weight, microbial growth, and in maintaining the water content of the fruit.

Consumer demand for eco-friendly and sustainable product packaging has proven to be remarkably stable and robust in recent years, including willingness to pay more for eco-friendly packaging. Consumers also recognize the value of reuse. The refillable packaging is proving to be a versatile and valuable solution for consumer products. Therefore, a holistic view of these issues is a growing requirement for everyone involved, from production, conservation, and marketing of this type of food product. Food choice is one of the most frequent human decisions and is determined by a complex set of factors and interrelated determinants [183]. Although several models attempting to explain that process have been proposed, one of the most accepted is the Total Food Quality Model [184]. This model can be divided into three parameters: ‘search’, ‘experience’, and ‘credence’ attributes. The first two (search attributes, such as appearance or price, and experience characteristics, such as flavor or taste) are those more easily observed by consumers and can be straightforwardly experienced by them. For credence properties, such as health and nutritional benefits, the consumer cannot validate those claims [185].

This is even more important in the current society, where the available fast food supply is large and more easily responds to the fast-paced life of consumers, with nutritionally poor foods taking place of a healthier diet.

## 5. Conclusions and Final Remarks

Nuts are a good source of many bioactive compounds with recognized health benefits, such as tocopherols, vitamins, and phenolic compounds. However, acquiring knowledge about the variation of bioactive compounds during fruit development and the ripening stage is crucial. How global environmental change and innovative crop production technology affect tree physiology and thus yield and fruit quality is at the moment mostly unknown. The development of species-specific strategies that improve both fruit quality and nutritional properties without significantly affecting yield should be aimed at by future research studies. The selection of high-yielding nut species and cultivars well-adapted to the different growing regions and future climatic conditions, with improved fruit traits, are needed to produce fruits with excellent quality and high consumer acceptability.

## Figures and Tables

**Figure 1 foods-12-00942-f001:**
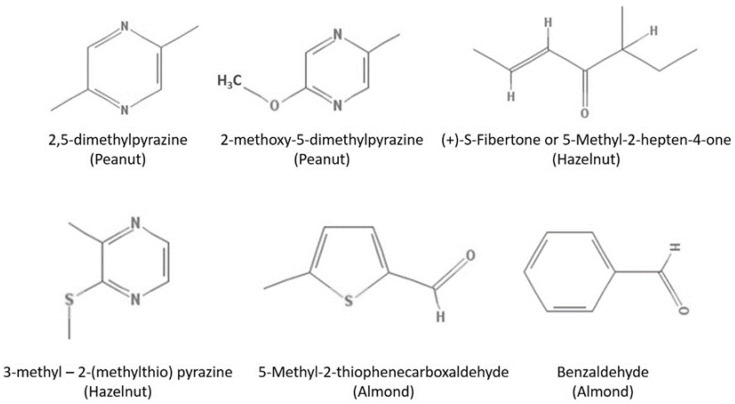
Some representative flavor compounds in almonds, peanuts, and hazelnuts.

**Table 1 foods-12-00942-t001:** Protein and amino acid contents of selected nuts (compiled from [10]).

Nut	Protein (g/100 g)	Amino Acids (g/100 g of Portion)																	
		Trp	Thr	Ile	Leu	Lys	Met	Cys	Phe	Tyr	Val	Arg	His	Ala	Asp	Glu	Gly	Pro	Ser
Almond	16.8–25.4	0.211	0.601	0.751	1.47	0.568	0.157	0.215	1.13	0.45	0.855	2.46	0.539	0.999	2.64	6.21	1.43	0.969	0.912
Cashew nut	17.5–19.0	0.287	0.688	0.789	1.47	0.928	0.362	0.393	0.951	0.508	1.09	2.12	0.456	0.837	1.80	4.51	0.937	0.812	1.08
Chestnut	1.63	0.018	0.058	0.064	0.096	0.096	0.038	0.052	0.069	0.045	0.091	0.116	0.045	0.109	0.028	0.021	0,084	0,086	0.081
Hazelnut	14.5–15.2	0.193	0.497	0.545	1.06	0.42	0.221	0.277	0.663	0.362	0.701	2.21	0.432	0.73	1.68	3.71	0.724	0.561	0.735
Macadamia nut	7.55–8.58	0.067	0.37	0.314	0.602	0.018	0.023	0.006	0.665	0.511	0.363	1.40	0.195	0.388	1.10	2.27	0.454	0.468	0.419
Peanut	25.8	0.25	0.883	0.907	1.67	0.926	0.317	0.331	1.38	1.05	1.08	3.08	0.652	1.02	3.15	5.39	1.55	1.14	1.27
Pecan nut	9.0–9.3	0.093	0.306	0.336	0.598	0.287	0.183	0.152	0.426	0.215	0.411	1.18	0.262	0.397	0.929	1.83	0.453	0.363	0.474
Pine nut	13.7	0.107	0.37	0.542	0.991	0.54	0.259	0.289	0.524	0.509	0.687	2.41	0.341	0.684	1.3	2.93	0.691	0.673	0.835
Pistachio	19.4–22.1	0.251	0.684	0.917	1.60	1.14	0.36	0.292	1.09	0.509	1.25	2.13	0.512	0.973	1.88	4.3	1.01	0.938	1.28
Walnut	14.4–16.0	0.17	0.596	0.625	1.17	0.424	0.236	0.208	0.711	0.406	0.753	2.28	0.391	0.696	1.83	2.82	0.816	0.706	0.934

Trp—Tryptophan, Thr—Threonine, Ile—Isoleucine, Leu—Leucine, Lys—Lysine, Met—Methionine, Cys—Cystine, Phe—Phenylalanine, Tyr—Tyrosine, Val—Valine, Arg—Arginine, His—Histidine, Ala—Alanine, Asp—Aspartic Acid, Glu—Glutamic Acid, Gly—Glycine, Pro—Proline, Ser—Serine.

**Table 2 foods-12-00942-t002:** Vitamin contents (mg/100 g) of selected nuts (source: [10,11,19,31,32,33]).

Nut	Ascorbic Acid (C)	Vit A (IU)	Niacin (B3)	Thiamine (B1)	Riboflavin (B2)	Pyridoxine (B6)	Folic Acid (B9)	Pantothenic Acid (B5)	α-Tocopherol (E)
Almond	3.62–3.90	0.06	3.62–3.90	0.21	0.80–1.14	0.1	0.04	0.3	2.4–25.9
Cashew nut	1.06–1.10	-	1.06–1.10	0.42	0.06–0.10	0.4	0.25	0.9	0.0–0.9
Hazelnut	1.81	20	1.81	0.30	0.10	0.2–0.6	ND	0.9	3.5–15.0
Peanut	5.75–12.10	-	5.75–12.10	0.60	0.04–0.10	0.1–0.3	0.24	0.6	0.4
Pine nut	4.40	29	4.40		0.20	0.1	ND	0.3	2.5–9.3
Pistachio	1.30	415	1.30	0.87	0.16–0.20	1.7	51.00	0.5	0.3–2.3
Walnut	0.47–1.13	20	0.47–1.13	0.34	0.15–0.20	0.5–0.6	0.98	0.6	0.1–13.0
Chestnut	40.2	26	1.1	0.14	0.02	ND	58	0.48	-

Not detected—ND.

**Table 3 foods-12-00942-t003:** Mineral contents (mg/100 g) of selected nuts (source: [10,11,19,31,32,33,35,36,37]).

Nut	Na	Mg	K	Ca	Cu	Zn	Fe
Almond	1.00	275	728	248	0.90–1.03	1.91–3.12	3.71–6.21
Cashew nut	12.00	292	660	37	0.56	0.96–5.78	3.82–6.68
Chestnut	2.00	30.00	484.00	19.00	0.418	0.49	0.94
Hazelnut	0.70–0.98	140–163	514–680	84–114	0.65–0.99	1.95–2.96	0.56–4.70
Peanut	1.30–18.00	168–173	558–705	67–92	0.75–0.83	0.44–3.27	0.58–4.58
Pine nut	2.00	251–265	597	16	1.32–1.60	3.08–6.45	5.53–6.64
Pistachio	1.00–9.36	117 -121	642–1025	107–171	0.75–1.70	2.77–6.72	0.41–8.86
Walnut	2.00	158–201	441–523	61–98	2.54	1.52–3.37	2.91–5.74

**Table 4 foods-12-00942-t004:** Fiber and lipid contents (%) of selected nuts (adapted from Amarowicz et al. [68]).

Nut	Fiber (%)	Lipid (%)
Almond	11.8–13.0	43.3–50.6
Cashew nut	1.4–3.3	42.8–43.9
Chestnut	2.3–3.7	1.6–7.4
Hazelnut	3.4–9.7	59.8–61.5
Pistachio	10.3	44.4–45.4
Walnut	6.7	64.5–65.2

**Table 5 foods-12-00942-t005:** Fatty acid composition of selected nuts (g/100 g nut) (Source: [10]).

Nut	SFA			MUFA			PUFA		
	Total	Palmitic 16:0	Stearic 18:0	Total	Oleic 18:1	Palmitoleic 16:1	Total	Linoleic 18:2	Linolenic 18:3
Almond	3.802	3.083	0.704	31.551	31.294	0.227	12.329	12.324	0.003
Cashew	7.783	3.916	3.223	23.797	23.523	0.136	7.845	7.782	0.062
Chestnut	0.425	0.384	0.021	0.780	0.749	0.021	0.894	0.798	0.095
Hazelnut	4.464	3.097	1.265	45.652	45.405	0.116	7.920	7.833	0.087
Pistachio	5.907	5.265	0.478	23.257	22.674	0.495	14.380	14.091	0.289
Walnut	6.126	4.404	1.659	8.933	8.799	0.134 (C_20:1_)	47.174	38.093	9.080

**Table 6 foods-12-00942-t006:** Main phenolic compounds found in the most common nuts (skins + kernels).

Nut	Phenolic Compound	Reference
Almond	Catechin, epicatechin, naringenin, eriodictyol, gallic acid, caffeic acid, chlorogenic acid, *o*-coumaric acid, *p*-coumaric acid ferulic acid, hydroxybenzoic acid, protocatechuic, vanillic acid, quercetin, kaempferol, isorhamnetin	[71,108,116,117]
Chestnut	Gallic acid, syringic acid, chlorogenic acid, ferulic acid, vanillic acid, catechin, naringin, quercetin, ellagic acid	[107,118,119]
Hazelnut	Gallic acid, protocatechuic acid, caffeic acid, *o*-coumaric acid, *p*-coumaric acid, ferullic acid, catechin, epicatechin, epicatechin gallate, rutin	[116,117,120,121,122,123]
Peanut	Catechin, epicatechin, quercetin, isorhamnetin, gallic acid, protocatechuic, caffeic acid, *p*-coumaric acid, procyanidins A and B, trimers and tetramers, prodelphinidin	[109,117,124,125,126]
Pecan nut	Ellagic acid, catechin, gallic acid, hydroxybenzoic acid, *trans*-cinnamic acid, syringic acid, caffeic acid, *p*-coumaric acid, ferulic acid, naringenin, apigenin, quercetin, rutin, kaempferol, isorhamnetin, resveratrol	[118]
Pistachio	Cyanidin, gallic acid, protocatechuic, eriodictyol, catechin, epicatechin, epicatechin gallate, luteolin, quercetin, myricetin, procyanidin B1, trimers, and tetramers	[109,117,127]
Walnut	Vanillic acid, catechin, pyrocatechin, protocatechuic acid, epicatechin, syringic acid, gallic acid, juglone and cinnamic acid, ellagic acid, rutin	[118,128,129]

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
