# Peer review of "Composition of Nuts and Their Potential Health Benefits—An Overview"

_foods, 2023, doi:10.3390/foods12050942_

Round 1

Reviewer 1 Report

This manuscript analyzed the composition of some selected nuts and their potential health benefits. The results showed that nuts were rich in bioactive compounds such as tocopherols, vitamins, and phenolic compounds, with well-demonstrated potential health benefits. This manuscript was helpful to understand nuts and then choose them scientifically and reasonably. Maybe it was better to add some contents about the effects of processing and storage on the quality of nuts and give some suggestions for people how to choose them scientifically and reasonably.

Author Response

Review 1

Comments and Suggestions for Authors

This manuscript analyzed the composition of some selected nuts and their potential health benefits. The results showed that nuts were rich in bioactive compounds such as tocopherols, vitamins, and phenolic compounds, with well-demonstrated potential health benefits. This manuscript was helpful to understand nuts and then choose them scientifically and reasonably. Maybe it was better to add some contents about the effects of processing and storage on the quality of nuts and give some suggestions for people how to choose them scientifically and reasonably.

Authors – First we would like to thank reviewer 1 for the comments, suggestions and recommendations, which allowed improving our work. We have proceeded according to the instructions of the reviewer. Please see the statements added (lines 307-333).

‘Major concerns of the food industry related to the production and commercialization of nuts are the effects of processing and storage on the quality of nuts. Indeed, both temperature and humidity after harvesting can influence the appearance, moisture content, texture and sensory characteristics of nuts [149]. Specifically, higher post-harvest temperature conditions can reduce crispness, increase moisture content and change oiliness and sweetness, resulting in the development of rancidity [150]. According to Mexis et al. [151], the alteration of sensorial characteristics leads to the formation of unpleasant flavors in pistachios, almonds, peanuts and walnuts, as a result of alterations in the oxidation rate caused by high storage temperatures. Indeed, it is mentioned in this study that storage temperatures of 30, 36 and 40 °C showed that nuts are more rancid compared to those stored at 8, 10, 20 or 25 °C.

Another possible alternative to increase the shelf life of nuts is the use of suitable packaging to reduce the problems mentioned above. Food can be packaged properly using modified atmosphere packaging or vacuum packaging to control the oxidation reaction [152]. The packaging material will be an important aspect to take into account as it will influence the shelf life of the nuts, as it will affect the respiration and transpiration rates of the fruit, as well as the development of microorganisms. Fernandes et al. [152], in their comparison studies of chestnut conservation packages, concluded that chestnut conservation through the use of a specific package can have a substantial impact on preserving the color and texture of the fruit, preventing the loss of weight, microbial growth, and in maintaining the water content of the fruit.

Consumer demand for eco-friendly and sustainable product packaging has proven to be remarkably stable and robust in recent years, including willingness to pay more for eco-friendly packaging. Consumers also recognize the value of reuse. Refillable packaging is proving to be a versatile and valuable solution for consumer products. Therefore, a holistic view of these issues is a growing requirement for everyone involved, from production, conservation and marketing of this type of food product.’

Reviewer 2 Report

This review summarizes, describes, discusses and compares the composition of nuts previously reported in the literature as well as recent advances in knowledge about the health benefits provided by them. The manuscript is well written and structured and fits very well within the scope of the special issue "Bioactive and (Poly)Phenolic Compounds: Characterization & Properties into Food and Health Benefits". However, some points should be carefully reviewed:

(1) Since this review summarizes nuts composition and their potential health benefits, studies that assess the bioaccessibility and/or bioavailability of nutrients and minerals in these foods should be added to item 2.

(2) The discussion and review regarding the advances in strategies that have been applied to improve some nut quality factors, such as cultivation in supplemented soil and different food packaging, should be expanded.

Additionally, the following minimum revisions are required:

Line 67: Replace “100g” with “100 g”. Space between number and unit must be added. Please review the entire text.

Table 1: Remove references from the legend and add a column containing references to the table, as in Table 4.

Line 171 - 174: The sentence is not clear, rephrase it.

Line 206-207: Exemplify some of the most commonly used methods to extract and quantify phenolic compounds in nuts.

Author Response

Review 2

Comments and Suggestions for Authors

This review summarizes, describes, discusses and compares the composition of nuts previously reported in the literature as well as recent advances in knowledge about the health benefits provided by them. The manuscript is well written and structured and fits very well within the scope of the special issue "Bioactive and (Poly)Phenolic Compounds: Characterization & Properties into Food and Health Benefits". However, some points should be carefully reviewed:

  1. Since this review summarizes nuts composition and their potential health benefits, studies that assess the bioaccessibility and/or bioavailability of nutrients and minerals in these foods should be added to item 2.

Authors: We would like to thank reviewer 2 for the comments, suggestions and recommendations, which allowed improving our work.

We agree with the reviewer’s comment that the bioaccessibility and/or bioavailability of nutrients and minerals of nuts are very important and should be explored in a review paper. However, currently there are hundreds of papers about this subject that which makes its introduction in this review impossible.

  1. The discussion and review regarding the advances in strategies that have been applied to improve some nut quality factors, such as cultivation in supplemented soil and different food packaging, should be expanded.

AuthorsWe appreciate the reviewer’s comment and we have proceeded according to the instructions of the reviewer. Please see the introduced text (lines 304-334)

‘Major concerns of the food industry related to the production and commercialization of nuts are the effects of processing and storage on the quality of nuts. Indeed, both temperature and humidity after harvesting can influence the appearance, moisture content, texture and sensory characteristics of nuts [149]. Specifically, higher post-harvest temperature conditions can reduce crispness, increase moisture content and change oiliness and sweetness, resulting in the development of rancidity [150]. According to Mexis et al. [151], the alteration of sensorial characteristics leads to the formation of unpleasant flavors in pistachios, almonds, peanuts and walnuts, as a result of alterations in the oxidation rate caused by high storage temperatures. Indeed, it is mentioned in this study that storage temperatures of 30, 36 and 40 °C showed that nuts are more rancid compared to those stored at 8, 10, 20 or 25 °C.

Another possible alternative to increase the shelf life of nuts is the use of suitable packaging to reduce the problems mentioned above. Food can be packaged properly using modified atmosphere packaging or vacuum packaging to control the oxidation reaction [152]. The packaging material will be an important aspect to take into account as it will influence the shelf life of the nuts, as it will affect the respiration and transpiration rates of the fruit, as well as the development of microorganisms. Fernandes et al. [152], in their comparison studies of chestnut conservation packages, concluded that chestnut conservation through the use of a specific package can have a substantial impact on preserving the color and texture of the fruit, preventing the loss of weight, microbial growth, and in maintaining the water content of the fruit.

Consumer demand for eco-friendly and sustainable product packaging has proven to be remarkably stable and robust in recent years, including willingness to pay more for eco-friendly packaging. Consumers also recognize the value of reuse. Refillable packaging is proving to be a versatile and valuable solution for consumer products. Therefore, a holistic view of these issues is a growing requirement for everyone involved, from production, conservation and marketing of this type of food product’.

Additionally, the following minimum revisions are required:

Line 67: Replace “100g” with “100 g”. Space between number and unit must be added. Please review the entire text.

Authors: Done.

Table 1: Remove references from the legend and add a column containing references to the table, as in Table 4.

Authors: We appreciate the comment, but as the Table is constructed is not possible to introduce the references by lines.

Line 171 - 174: The sentence is not clear, rephrase it.

Authors: Done.

Line 206-207: Exemplify some of the most commonly used methods to extract and quantify phenolic compounds in nuts.

Authors: The statement was reformulated as suggested by the reviewer. Please see the reformulation of the statement (Lines 201-205).

‘In addition, differences in the methods used to extract and quantify phenolic compounds (e.g., Microwave assisted extraction – MAE; Supercritical CO2 extraction – SC-CO2; Enzyme assisted extraction – EAE; and Pressurized liquid extraction – PLE) by researchers may interfere with the number of phenolic compounds identified. However, based on the literature it is possible to find a more or less common pattern.’

Reviewer 3 Report

The manuscript foods-2071853 aims to summarize the current knowledge and describe the most recent research on the health benefits of selected nuts.

I consider the paper to be relevant to Foods, represents a great contribution to future readers, and is quite easy to read. In particular, I think the researchers made some good divisions of the topics to be covered. However, there are some minor issues that I think should be addressed before:

- Place units to each numerical value; i.e., 8 % to 22 %. Fix this throughout the text.

-L63-64: remove capital letters.

These are the only major issues:

- Report only significant figures throughout the text. Pay special attention to the values reported in the Tables, where there is a mixture of significant figures.

- The great majority of references are very old. If the manuscript is an overview, I believe that the manuscripts cited, in their great majority (minimum 80%), should be from the last 5 years.

Author Response

Review 3

I consider the paper to be relevant to Foods, represents a great contribution to future readers, and is quite easy to read. In particular, I think the researchers made some good divisions of the topics to be covered. However, there are some minor issues that I think should be addressed before publishing the paper:

Authors: First we would like to thank reviewer 3 for the comments, suggestions and recommendations, which allowed improving our work. We have proceeded according to the instructions of the reviewer.

- Place units to each numerical value; i.e., 8 % to 22 %. Fix this throughout the text.

Authors: Done

-L63-64: remove capital letters.

Authors: Done

These are the only major issues:

- Report only significant figures throughout the text. Pay special attention to the values reported in the Tables, where there is a mixture of significant figures.

- The great majority of references are very old. If the manuscript is an overview, I believe that the manuscripts cited, in their great majority (minimum 80%), should be from the last 5 years.

Authors: We appreciated the reviewer’s comments and we had introduced more recent references.

Reviewer 4 Report

The manuscript entitled “Nuts composition and their potential health benefits-an overview” by Berta Conqalves et al is potentially interesting topic dealing with the health benefit and nuts.

General comments

1, The manuscript does not describe the characteristics of each nut precisely but describe the rough explanation of each nut.

2, The authors quote many references, however, they don’t explain enough each reference in the manuscript.

Specific comments

1, In table 1, the authors showed the content (amount) of vitamin B and vitamin E (VE) in each nut. The amount is different in each nut. For example, in the case of VE, almond, hazel nut, and pine nut contained VE almost ten times bigger than those of cashew, peanut, pistachio and walnut.  There is no clear explanation for this difference.  In the case of Na, cashew contained a large amount of Na. The blood pressure and Na ion is strongly related, it is important for this fact and health benefit.

2, In table 2, the amount of fiber and lipid content is shown, there is a big difference of the values written in the table. For example, almond and pistachio showed large value of fiber and lipid, while, chestnut showed low value of fiber and lipid. In the case of hazelnut, the fiber value is big, however, lipid value is low.

It is not clear for all these values and health benefit.

3, In table 3, authors showed composition of the fatty acid in the nut. The composition is heavily dependent on the nut.  It is also important to explain the difference of nut and health benefit.

4, In table 4, the authors showed the polyphenol compounds included in each nut without any figures of polyphenols. In a nutshell, polyphenol, there are many differences. For example, authors should be careful for the structure of each polyphenol. There is a big structural difference between gallic acid and epicatechin. If authors draw the structure of the compounds in table 4, the authors can understand the fact.

5, In figure 1, it is not meaningful to write these compounds. As far as writing something like this, the authors draw the key compounds in table 4.

6, In figure 2, the authors must improve this figure, if they want to summarize the manuscript.  This figure draws just a list of facts, it is better to draw the figure to clarify the mutual relationship and positioning of each factor.

Author Response

Review 4

The manuscript entitled “Nuts composition and their potential health benefits-an overview” by Berta Gonçalves et al is potentially interesting topic dealing with the health benefit and nuts.

 Authors: We would like to thank reviewer 4 for the comments, suggestions and recommendations, which allowed improving our work. We have proceeded according to the instructions of the reviewer

General comments

1, The manuscript does not describe the characteristics of each nut precisely but describe the rough explanation of each nut.

2, The authors quote many references, however, they don’t explain enough each reference in the manuscript.

Specific comments

1, In table 1, the authors showed the content (amount) of vitamin B and vitamin E (VE) in each nut. The amount is different in each nut. For example, in the case of VE, almond, hazel nut, and pine nut contained VE almost ten times bigger than those of cashew, peanut, pistachio and walnut.  There is no clear explanation for this difference.  In the case of Na, cashew contained a large amount of Na. The blood pressure and Na ion is strongly related, it is important for this fact and health benefit.

2, In table 2, the amount of fiber and lipid content is shown, there is a big difference of the values written in the table. For example, almond and pistachio showed large value of fiber and lipid, while, chestnut showed low value of fiber and lipid. In the case of hazelnut, the fiber value is big, however, lipid value is low.

It is not clear for all these values and health benefit.

3, In table 3, authors showed composition of the fatty acid in the nut. The composition is heavily dependent on the nut.  It is also important to explain the difference of nut and health benefit.

4, In table 4, the authors showed the polyphenol compounds included in each nut without any figures of polyphenols. In a nutshell, polyphenol, there are many differences. For example, authors should be careful for the structure of each polyphenol. There is a big structural difference between gallic acid and epicatechin. If authors draw the structure of the compounds in table 4, the authors can understand the fact.

Authors: We appreciate the reviewer’s comments and we agree with them. Indeed, it is a big challenge to all the researchers that work with fruit trees, because especially with nut trees the variation in minerals, lipids, vitamins, phenolics, etc. is very depending on the genotype, rootstock, edaphoclimatic conditions, climate change, crop technology, postharvest storage, among others (Please see Figure 2).

This review is focused on the research of many authors that analyses nut tree growing in many different regions in the world, therefore a big variation in the composition is expected. Here, we try to compilated the information available until the present.

5, In figure 1, it is not meaningful to write these compounds. As far as writing something like this, the authors draw the key compounds in table 4.

Authors: We appreciate the reviewer’s comment. Our objective was slightly present some flavor compounds in nuts, since this review is about the major compounds that have high potential health benefits.

6, In figure 2, the authors must improve this figure, if they want to summarize the manuscript.  This figure draws just a list of facts, it is better to draw the figure to clarify the mutual relationship and positioning of each factor.

Authors: We appreciate the reviewer’s comment and we agree that it is a very simple Figure. The objective is to highlight the most important aspects regarding health benefits for the consumers when they eat nuts.

Reviewer 5 Report

The manuscript by Gonçalves and Co-authors aims to summarize current knowledge and describe the most recent research regarding the health benefits of some selected nuts.

The review is well-written and well-organized. 

However, my concerns arose from the very low degree of novelty. The potential health benefits of nuts and their chemical composition are well described and reported in the literature, and several reviews are available. Thus, to be accepted, the authors should highlight and stress what makes this review different from those already available 

Author Response

Review5

Comments and Suggestions for Authors

The manuscript by Gonçalves and Co-authors aims to summarize current knowledge and describe the most recent research regarding the health benefits of some selected nuts.

The review is well-written and well-organized. 

However, my concerns arose from the very low degree of novelty. The potential health benefits of nuts and their chemical composition are well described and reported in the literature, and several reviews are available. Thus, to be accepted, the authors should highlight and stress what makes this review different from those already available 

Authors: We deeply appreciated the positive comments on our manuscript. This review is focused on the research of many authors that analyses nut tree growing in many different regions in the world, therefore a big variation in the composition is expected. Here, we try to compilated the information available until the present. So, this overview aims to summarize current knowledge and describe the most recent research regarding the health benefits of some selected nuts.

Round 2

Reviewer 3 Report

The authors do not follow the major recommendations.

Author Response

Response to Reviewers

Reviewer #3

The authors do not follow the major recommendations.

Authors: Thank you for giving us the opportunity to clarify the changes made in the manuscript based on your recommendations. All changes are highlighted in yellow.

- Place units to each numerical value; i.e., 8 % to 22 %. Fix this throughout the text.

Authors: We deeply apology for this error and corrected it throughout the text. Thank you.

-L63-64: remove capital letters.

Authors: Done

These are the only major issues:

- Report only significant figures throughout the text. Pay special attention to the values reported in the Tables, where there is a mixture of significant figures.

Authors: You are right. We revised all text and Tables and we used the same significant digits within each component.

- The great majority of references are very old. If the manuscript is an overview, I believe that the manuscripts cited, in their great majority (minimum 80%), should be from the last 5 years.

Authors: We agree with your suggestion that more recent references need to be added. Our purpose is to provide a broad compilation of the information available until the present and, as such, old references should be referred. When possible, we incorporated more recent references. Following your suggestion, we have therefore added these references:

Shirmohammadi, M.; Chandrasekaran, I.; Singh, C. Effect of post-harvest processes and storage conditions on aging and quality of fruit nuts. Annual Conference, School of Engineering, 2019, University of Guelph, Guelph, Ontario. https://www.researchgate.net/publication/335160450

Kader, A.A. Impact of nut postharvest handling, deshelling, drying and storage on quality. In Improving the safety and quality of nuts, Harris L. J., Ed.; Publisher: Woodhead Publishing Ltd Cambridge, UK, 2013; Volume 1, pp. 22–34.

Mexis, S.F.; Badeka, A.V.; Riganakos, K.A.; Karakostas, K.X.; Kontominas, M.G. Effect of packaging and storage conditions on quality of shelled walnuts. Food Control 2009, 20(8), 743-751. doi: 10.1016/j.foodcont.2008.09.022

Fernandes, L.; Pereira, E.L.; Fidalgo, M.C.; Gomes, A.; Ramalhosa, E. Effect of modified atmosphere, vacuum and polyethylene packaging on physicochemical and microbial quality of chestnuts (Castanea sativa) during storage. Int. J. Fruit Sci. 2020, 20(2), S785-S801. doi: 10.1080/15538362.2020.1768619

Reviewer 4 Report

The authors did not answer clearly the points which the reviewer indicated.

Please send the manuscript well-organized form and revised points should be shown clearly one by one form.

Author Response

Reviewer #4

The authors did not answer clearly the points which the reviewer indicated.

Please send the manuscript well-organized form and revised points should be shown clearly one by one form.

Authors: We deeply apologize for the lack of clarity. All comments from reviewer #4 are addressed below after each point.

General comments

1, The manuscript does not describe the characteristics of each nut precisely but describe the rough explanation of each nut.

Authors: We agree with this comment. Our main attempt was to compile the available data about the composition of the most consumed nuts in the world. Taking into consideration, the wide variety of nuts and their cultivars, a precise description of each one is not feasible.

2, The authors quote many references, however, they don’t explain enough each reference in the manuscript.

Authors: We used several references and all of them were mentioned in the text, mainly in the tables. It is thus impossible to explain in detail each reference.

Specific comments

1, In table 1, the authors showed the content (amount) of vitamin B and vitamin E (VE) in each nut. The amount is different in each nut. For example, in the case of VE, almond, hazel nut, and pine nut contained VE almost ten times bigger than those of cashew, peanut, pistachio and walnut. There is no clear explanation for this difference. In the case of Na, cashew contained a large amount of Na. The blood pressure and Na ion is strongly related, it is important for this fact and health benefit.

2, In table 2, the amount of fiber and lipid content is shown, there is a big difference of the values written in the table. For example, almond and pistachio showed large value of fiber and lipid, while, chestnut showed low value of fiber and lipid. In the case of hazelnut, the fiber value is big, however, lipid value is low.

It is not clear for all these values and health benefit.

3, In table 3, authors showed composition of the fatty acid in the nut. The composition is heavily dependent on the nut. It is also important to explain the difference of nut and health benefit.

Authors: Thank you for these comments (1, 2 & 3). It would have been interesting to explore these aspects. However, in this review, we tried to emphasize the general benefits of nut consumption instead of focusing on a specific nut species.

4, In table 4, the authors showed the polyphenol compounds included in each nut without any figures of polyphenols. In a nutshell, polyphenol, there are many differences. For example, authors should be careful for the structure of each polyphenol. There is a big structural difference between gallic acid and epicatechin. If authors draw the structure of the compounds in table 4, the authors can understand the fact.

Authors: We thank the suggestion, but we think this type of review does not justify the inclusion of such figures. Our focus is to summarize the major compounds that have high potential health benefits. Moreover, the variation in minerals, lipids, vitamins, phenolic, etc. is very depending on the genotype, rootstock, edaphoclimatic conditions, climate change, crop technology and postharvest storage, among others, which could accentuate or attenuate the differences among nuts.

5, In figure 1, it is not meaningful to write these compounds. As far as writing something like this, the authors draw the key compounds in table 4.

Authors: We understand your point, but we would prefer to keep this figure because this aspect (flavour compound) is less explored in the literature.

6, In figure 2, the authors must improve this figure, if they want to summarize the manuscript. This figure draws just a list of facts, it is better to draw the figure to clarify the mutual relationship and positioning of each factor.

Authors: We agree that it is a very simple Figure. The objective is to highlight the most important aspects regarding health benefits for the consumers when they eat nuts.

Reviewer 5 Report

The review can be accepted in the present form

Author Response

Reviewer #5

The review can be accepted in the present form

Authors: We thank you for the positive feedback.
